# An Analysis of the Trend of Fetal Mortality Rates among Working and Jobless Households in Japan, 1995–2019

**DOI:** 10.3390/ijerph18094810

**Published:** 2021-04-30

**Authors:** Tasuku Okui

**Affiliations:** Medical Information Center, Kyushu University Hospital, Fukuoka 812-8582, Japan; okui.tasuku.509@m.kyushu-u.ac.jp

**Keywords:** fetal mortality, Japan, induced abortion, spontaneous abortion, socioeconomic factors

## Abstract

This study aimed to identify differences in the trends of artificial and spontaneous fetal mortality rates between working and jobless households depending on ages, periods, and birth cohorts in Japan. Vital Statistics data from 1995 to 2019 and age groups in 5–year increments from 15 to 19 years through 45 to 49 years were used. Bayesian age–period–cohort analysis was used to evaluate changes in each of the outcomes. As a result, the difference in maternal age–standardized rate of both the artificial and spontaneous fetal mortality rates between the two types of households decreased in the periods analyzed. However, there was a statistically significant difference in the mortality rate between jobless and working households, regardless of maternal ages, periods, and cohorts for the artificial fetal mortality rate. A statistically significant difference was also observed for the spontaneous fetal mortality rates in some maternal ages, periods, and cohorts. In addition, the trend of birth cohort effects was particularly different between the two types of households for both the artificial and spontaneous fetal mortality rates.

## 1. Introduction

Japan is recognized as having favorable outcomes for maternal and child health in recent years [1]. The perinatal mortality rates and infant mortality rates in Japan are among the lowest in the world, with an infant mortality rate of 2.0 per 1000 births and a perinatal mortality rate of 3.6 per 1000 births in 2016 [1]. There are many indicators of maternal and child health; fetal mortality rate is a representative measure along with infant mortality rate and perinatal mortality rate. Fetal mortality includes mortality caused by spontaneous abortion and mortality caused by induced abortion, and the number of induced abortions is larger than that of spontaneous abortions in Japan [1]. Although induced abortion is legally permitted in Japan, the number of fetal mortality by induced abortions and spontaneous abortions has continued to decrease since the 1950s [1]. In addition, the trends of representative indicators of maternal and child health, including fetal mortality rates, have been analyzed using the Vital Statistics in Japan in recent years [2,3,4,5,6]. On the other hand, the trends of maternal age standardized values have not been revealed despite the fact that these indicators of maternal and child health are largely influenced by maternal age [7,8].

It is also known that socioeconomic status (SES) affects maternal and child health [9,10,11,12,13]. Regarding fetal mortality (including stillbirth), SES differences have been also observed in many countries [14,15,16,17,18,19,20]. Stillbirth, rather than fetal mortality, is often focused on in other countries, and disparities depending on races, educational levels, occupations, and incomes have been shown in previous studies [14,18,19,20]. In Japan, household occupation, which is available in the Vital Statistics, can be used as an indicator of SES in the analysis of the Vital Statistics, and an analysis of an association between household occupation and fetal mortality rates has been conducted [4]. The result indicated that jobless households were associated with higher fetal mortality rates than working households. However, the periods analyzed in the previous studies were 1995–2004, and the data of recent years have not been analyzed. In addition, the differences between working and jobless households based on age, period and cohort have not been analyzed. Fetal mortality rates change depending on ages and periods, and it is considered that the relationship between working and jobless households varies depending on these factors.

An age–period–cohort (APC) analysis is often conducted for analyzing the trend of incidence or mortality rate of a disease [21]. By using this analysis method, age, period and cohort effects for the trend of an outcome can be identified. Although some APC analyses for indicators of perinatal health have been conducted in other countries [22,23], an analysis for these indicators has not been conducted in Japan. By applying this analysis method for each type of household, the differences in the indicators between SES depending on age, period and cohort can be assessed, and the structure of the disparity can be better understood. In this study, an APC analysis was conducted for fetal mortality rate among working and jobless households in Japan using the Vital Statistics.

## 2. Materials and Methods

In Japan, delivery of a stillborn child after the 12th week of pregnancy is treated as fetal mortality, and data of artificial and spontaneous fetal mortality are publicly available in the Vital Statistics [24]. Artificial fetal mortality is fetal mortality caused by induced abortion after the 12th week of pregnancy. The artificial fetal mortality rate in this study is defined as artificial fetal mortality per the sum of the number of fetal mortality and the number of births. The artificial fetal mortality number indicates the number of fetal mortalities caused by induced abortion after the 12th week of pregnancy. On the other hand, spontaneous fetal mortality is fetal mortality caused by spontaneous abortion after the 12th week of pregnancy. The spontaneous fetal mortality rate is defined as number of fetal mortalities by spontaneous abortion per the sum of total number of births and fetal mortality. The definitions of fetal mortality rate in this study correspond to those used in the Vital Statistics [24]. In the case of multiple fetuses, the number of deceased fetuses is included in the total number of cases of fetal mortality in Japan. For example, when two fetuses in a quadruplet died, it is defined as two cases of fetal mortality.

The data of the Vital Statistics in Japan are from 1995 to 2019 [24]. Information for maternal age groups in 5–year increments from 15–19 years through 45–49 years were publicly available and used for the analysis. The fetal mortality data of maternal age groups under 15 years and above 50 years were excluded from the analysis because births given by women in these age groups were few in Japan. The age group of 45–49 years in 1995 was defined as the oldest birth cohort in the analysis data set. Through 1-year shifts starting from the oldest cohort, the age group of 15–19 years in 2019 was defined as the earliest cohort. Then the number of births, artificial fetal mortality, spontaneous fetal mortality for each type of household, maternal age group, and year were extracted. For all analyses, the births and mortalities whose household occupations were uncertain were excluded from the analysis. Jobless households were defined as households in which both parents were not working when fetal mortality occurred. In addition, a trend in the gross domestic product (GDP) is often associated with a trend of an indicator of child and maternal health [25]. Therefore, the trend of the GDP was also shown in the results. Data on the GDP were obtained from the Cabinet office [26], and the GDP per capita was calculated using total population in Japan [27].

For statistical analysis, the artificial and spontaneous fetal mortality rate during the analyzed periods for each age group and type of household were calculated. Moreover, maternal age-standardized rates for each of the indicators were calculated using the proportion of each age group for the total births in 1995 to 1999 as the standard population. The analyzed periods were aggregated into 5 periods in these descriptive analyses for visibility of the results, but the data of every single year were used in the APC analysis. In addition, yearly artificial and spontaneous fetal mortality rate per 1000 births and GDP per capita were calculated to verify the relationship. Then an APC analysis for each outcome and each type of household was conducted. The Bayesian APC model based on the Poisson regression model was used in our study [28]. The restriction that the sum of each APC effect is zero was applied in order to identify each effect. As the prior for each APC affect, random-walk of first-order was used. For estimating the parameters, Stan was used (http://mc-stan.org; accessed on 30 April 2021). The analysis was conducted for working and jobless households, separately. The estimated artificial fetal mortality rate for working households for each age group, period, and cohort using the estimated intercept and each effect was also calculated, as was often conducted in previous studies [29]. The artificial and spontaneous fetal mortality rate ratio of jobless households compared with working households was then calculated for each age group, period, and cohort. All the statistical analyses were conducted using R 3.6.3 software (https://www.r-project.org/; accessed on 30 April 2021).

## 3. Results

Table 1 indicates the number of artificial and spontaneous fetal mortalities and births in each of the periods and maternal ages. The number of artificial fetal mortalities was larger than that of spontaneous fetal mortality in all the age groups and periods. The number of artificial fetal mortalities decreased from 1995–1999 to 2015–2019 in all the age groups, regardless of the type of households. The number of spontaneous fetal mortalities also decreased over the years in many age groups, while an increase in the number was observed in older age groups for working households. Over the years, the number of births decreased in the age groups under 35 years, while it increased in the age groups of 35 years or more.

Table 2 shows the artificial and spontaneous fetal mortality rate per 1000 births for each maternal age group and the maternal age-standardized rate for each period and type of household. From this result, it was obvious that a disparity existed between the two types of households. The values of jobless households were higher than that of working households, regardless of age group, period, and the indicators in most of the cases. Artificial and spontaneous fetal mortality rates decreased in most of the age groups during the analyzed periods, regardless of the type of household. In addition, the difference between the two types of households decreased during the analyzed periods for both indicators.

The Appendix A shows the yearly artificial and spontaneous fetal mortality rates per 1000 births and GDP per capita in Japan. There was no obvious correlation or reverse correlation between the GDP per capita and yearly fetal mortality rates, and none of the mortality rates were correlated with the GDP per capita based on the Spearman’s rank correlation coefficient.

Figure 1 shows the age, period and cohort effect for the artificial and spontaneous fetal mortality rate among jobless and working households. Mortality rates changed depending on maternal ages, periods, and cohorts regardless of the types of households. The maternal age effect became the largest in the age group from 15–19 years old, regardless of the type of household for artificial fetal mortality rate, and the period effects showed decreasing trends for both types of households. Moreover, the cohort effect for working households decreased until cohorts born in approximately 1970 and showed subtle increasing trends in cohorts born more recently for artificial fetal mortality rate. Maternal age effects for the spontaneous fetal mortality rate became larger with an increase in age for both types of households. Although the period effects for spontaneous fetal mortality rate decreased for both types of households, the degree of the decrease was larger for jobless households. On the other hand, the degree of the decrease of the cohort effect was larger for working households.

Figure 2 shows the rate ratios of jobless households compared with working households for the artificial and spontaneous fetal mortality rate. It was shown that the degree of the disparity changed depending on maternal ages, periods, and cohorts. There was a statistically significant difference between the two types of households in all ages, periods, and cohorts for the artificial fetal mortality rate. The artificial fetal mortality rate ratio became the largest among the group aged 25–29 years old. Moreover, although the rate ratio increased until cohorts born in the 1970s, it showed a decreasing trend thereafter. Regarding the spontaneous fetal mortality rate, estimates of the rate ratio were above 1 in many of the cases. A statistically significant difference between the type of households was observed in some of the ages, periods, and cohorts. The rate ratio showed a decreasing trend with an increase in age for spontaneous fetal mortality rate. Although the spontaneous fetal mortality rate ratio showed a decreasing trend during the analyzed period, it showed an increasing trend from cohorts born in the late 1940s to the 1970s.

## 4. Discussion

In regard to the artificial fetal mortality rate, the degree of the differences between the two types of households was larger than the spontaneous fetal mortality rate. It is known that SES is associated with the rate of induced abortion in Japan and other countries [4,30,31], and the results of this study demonstrated that the degree of disparity between the two types of households changed significantly depending on maternal ages and cohorts. According to a previous study, approximately 40% of the pregnancies are unexpected pregnancies in Japan [32]. In addition, it is known that the estimated number of unexpected pregnancies in 2016 was 610,000 [33], which was much larger than the number of induced abortions. Therefore, a higher percentage of unexpected pregnancies have been considered to result in childbirth rather than abortion, and it was suggested that the type of households may have affected the induced abortion rates during unexpected pregnancies. The age effect was the largest in the age group between 15–19 years old, regardless of the type of household. According to a previous study of induced abortion in Japan, the induced abortion rate in those between 40–44 years old was the highest from 1980 to 1995 [34]. The induced abortion in the previous study included not only artificial fetal mortality, but also induced abortion conducted before the 12th week of pregnancy [34], and it is considered that the proportion of induced abortion after the 12th week of pregnancy is particularly high in the age group between 15–19 years old in Japan. Moreover, in working households, the artificial fetal mortality rate in the age group between 15–19 years old was particularly higher than in the other age groups. The rate of sexual experience in younger ages increased beginning in the late 20th century, and this is said to be a factor for the high rate of induced abortion among younger age groups [34]. It is also pointed out that knowledge of contraceptive methods is insufficient in Japan [34] and that an effective contraceptive method is not widespread in Japan [33], and there is a possibility that these factors contributed to the higher artificial fetal mortality rate in the younger age groups. With regard to the period effects, the rate ratio decreased until the early 2000s, because the period effect increased for working households, as shown in Figure 1. A previous study demonstrated that rate of induced abortion in Japan increased from 1998 to 2001 and that changes in sexual behavior and the use of contraceptives are believed to be factors [35]. On the other hand, it is known that the proportion of young Japanese adults with no experience of heterosexual intercourse has increased in the past decades [36], and it might have contributed to the decline of the period effects for both types of households. With regard to the trend of the cohort effect, the trends were largely different depending on the type of household. The rate of induced abortion largely decreased beginning at the end of World War II in Japan, possibly resulting from the decrease of unexpected pregnancy [34,35]. A previous study indicated that the transition from induced abortion to contraception began in the 1950s and ended in the 1970s [37]. From the results of this study, it is considered that contraception has spread among working households, but not among jobless households. Jobless households are highly associated with a low educational background in Japan [38], and this might have affected the differences in the spread of contraception use and behaviors. On the other hand, the artificial fetal mortality rate of working households began to increase from cohorts born in the 1970s, and the difference between the households was shrinking over the cohorts. Therefore, there is a possibility that the rate of unexpected pregnancy in working households was increasing over the cohorts. Taking into account the decrease of experience of heterosexual intercourse in the past decades [36], inappropriate contraceptive behaviors might have increased over the cohorts.

Spontaneous fetal mortality rates increased with maternal age for both types of households, and the degree of increase of the age effect was larger for working households. Maternal age is a known major risk factor for stillbirths [7,8,39], but the disparity between the two types of household decreased in older ages for which the risk of perinatal mortality was high. The period effects decreased in both types of households, and the degree of the decrease was rather larger in jobless households. On the other hand, reverse trends were observed in the cohort effects. The improvement of accessibility to perinatal care centers and public health interventions focusing on maternal and child health are considered to be major factors for the decline of the perinatal mortality rate in Japan [40,41], and it is considered to be also related to the result of spontaneous fetal mortality rates. In addition, public subsidies for prenatal care were implemented in 2009 and are said to be effective for women who cannot receive the care for financial reasons [41,42]. Therefore, it is believed that the advances in perinatal care and prenatal check-ups were more effective for jobless households in the periods. In the analyzed periods, there also were some huge social events in Japan, and the financial crisis and the Great East Japan earthquake occurred in 2007–2008 and in 2011, respectively. It is known that the global financial crisis predisposed to low birth weight in Portugal and Greece [43,44]; however, no apparent negative effect on perinatal outcomes was observed for women who experienced the earthquake in Japan [45]. Actually, there were no obvious changes in the trend of period effects at the time of these events. In regard to the cohort effects, a change in the cohort effect reflects a change in the maternal factors over the cohorts, and there is a possibility that it reflects the change of fetal mortality caused by perinatal pathologies, such as pre-eclampsia. However, a large proportion of spontaneous fetal mortality occurs in early pregnancy [24], and other factors in addition to perinatal diseases are considered to be related to the result. A factor that can change depending on maternal birth cohorts is educational level. Low educational level is a risk factor for stillbirth [13], and educational level improved over birth cohorts in Japan [46]. Therefore, an increase in the spontaneous fetal mortality rate ratio from cohorts born in the late 1940s to the 1970s between the two types of workers might indicate the difference in trends of maternal educational level, and it applies to the results of the artificial fetal mortality rate.

Fertility decline is one of the biggest social problems in Japan, and the government has taken numerous measures against this problem over decades [47]. On the other hand, there have been hundreds of thousands of cases of fetal mortality over the decades, as shown in Table 1. The disparity shown in this study implies that a certain proportion of artificial fetal mortality occurred because of socioeconomic reasons. Therefore, it is considered that revealing the nature of disparity in the fetal mortality rates can aid in identifying a target population that needs financial aids for childbirth.

This study has several limitations. First, an APC analysis is a type of descriptive analysis method, and the accurate reason for the differences in the trend of each APC effect between the two types of households is uncertain. Other factors, such as maternal physical characteristics or prevalence of complications of pregnancy, are also considered to be related to the differences. For example, gestational age and experience of past births are known to affect perinatal outcomes [9,17,48], and there might exist some differences between the two types of households. Studies focusing on physical and SES characteristics of mothers in jobless households need to be conducted to reveal the reasons for the differences between the two types of households. In addition, although this study focused on households with or without jobs as the SES indicator, differences in the outcomes for maternal and child health might also be evident among income levels of households or educational level. It will be meaningful to conduct an epidemiological study focusing on other SES of the households in the future.

## 5. Conclusions

The difference in the maternal age-standardized rate of both the artificial and spontaneous fetal mortality rates between the two types of households decreased in the analyzed periods. However, there was a statistically significant difference in the rate ratio of jobless households compared with working households, regardless of maternal ages, periods, and cohorts for the artificial fetal mortality rate. A statistically significant difference was also observed for the spontaneous fetal mortality rates in some ages, periods, and cohorts. Trends of birth cohort effects were particularly different between the two types of households for both the artificial and spontaneous fetal mortality rates. A large decrease of the cohort effect was observed from cohorts born in the late 1940s to the 1970s in working households, whereas it was not observed in jobless households. Further studies investigating the physical and socioeconomic characteristics of pregnant women for both jobless and working households are required for a better understanding of the reason for the disparity.

## Figures and Tables

**Figure 1 ijerph-18-04810-f001:**
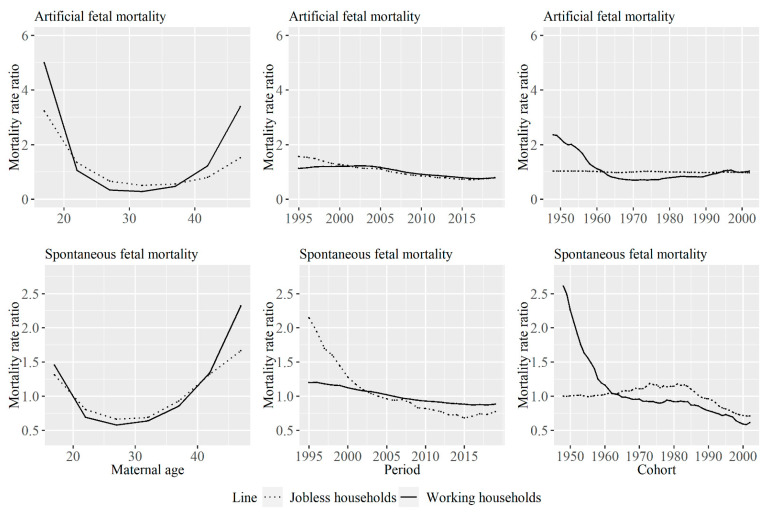
The age, period, and cohort effect for the artificial and spontaneous fetal mortality rate among jobless and working households. The graph shows the age, period, and cohort effects for induced abortion rate and perinatal mortality rate for each type of household. Solid lines signify estimates of each effect for working households, and dotted lines signify estimates of each effect for jobless households.

**Figure 2 ijerph-18-04810-f002:**
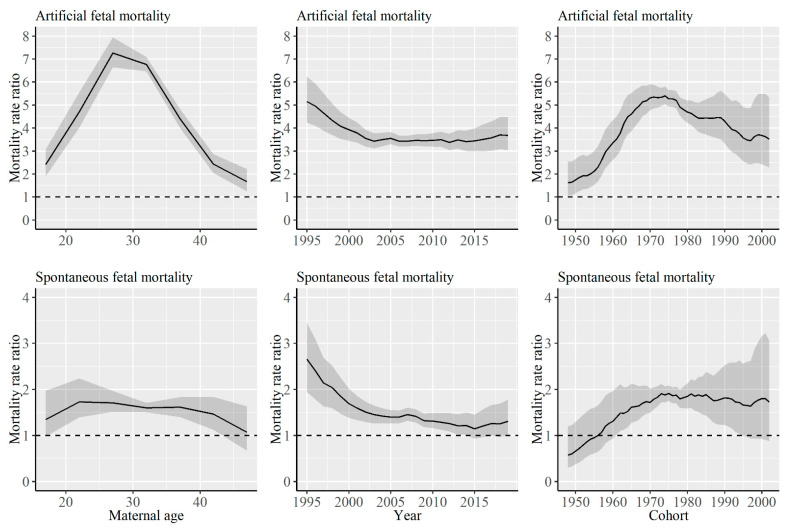
The rate ratios of jobless households compared with working households for the artificial and spontaneous mortality rate. Lines signify estimates of the rate ratios of jobless households compared with working households, and the shadings denote the 95% credible intervals of the rate ratio.

**Table 1 ijerph-18-04810-t001:** The number of artificial and spontaneous fetal mortalities and births in each of the periods and maternal ages.

Type of Indicators and Households	Maternal Age Group						
	15–19	20–24	25–29	30–34	35–39	40–44	45–49
Artificial fetal mortality (Jobless households)							
1995–1999	6375	6423	3271	1561	905	364	40
2000–2004	6785	5683	3231	2194	1090	371	34
2005–2009	3727	4012	2243	1756	1085	340	22
2010–2014	3085	2882	1599	1133	907	349	31
2015–2019	1883	2156	1038	721	502	274	28
Artificial fetal mortality (Working households)							
1995–1999	13,434	23,447	18,369	12,535	9209	5120	712
2000–2004	14,749	19,338	16,741	13,694	8822	4104	537
2005–2009	8547	14,546	12,380	12,145	9273	3675	361
2010–2014	7175	10,348	9852	9738	9149	4181	301
2015–2019	4819	9073	7836	8707	7999	4405	393
Spontaneous fetal mortality (Jobless households)							
1995–1999	949	1357	1149	776	454	178	13
2000–2004	786	999	912	749	463	155	5
2005–2009	352	634	604	657	498	141	10
2010–2014	239	427	451	427	430	180	13
2015–2019	131	267	265	286	263	124	7
Spontaneous fetal mortality (Working households)							
1995–1999	2591	12,126	27,337	24,440	10,710	2731	249
2000–2004	2232	8301	20,902	23,244	11,001	2422	161
2005–2009	1122	6034	14,602	20,637	12,868	2884	120
2010–2014	821	4106	11,652	17,291	13,997	4203	155
2015–2019	535	3038	8864	15,216	12,853	4470	204
Number of births (Jobless households)							
1995–1999	5807	22,088	28,567	20,295	8915	2204	122
2000–2004	9123	28,215	36,227	31,503	14,361	3041	116
2005–2009	7696	25,761	29,179	28,522	16,904	3834	136
2010–2014	8010	23,281	26,749	24,390	17,275	5010	204
2015–2019	5842	18,331	17,781	17,712	12,582	4109	167
Number of births (Working households)							
1995–1999	75,960	872,170	2,399,876	1,855,720	528,206	61,555	1945
2000–2004	85,803	696,698	2,025,668	1,950,788	644,886	77,215	1898
2005–2009	63,983	569,571	1,535,287	1,935,638	873,436	115,753	2693
2010–2014	52,029	441,545	1,372,134	1,768,699	1,070,946	198,190	4377
2015–2019	39,746	358,983	1,152,016	1,645,137	1,036,668	246,721	6817

**Table 2 ijerph-18-04810-t002:** The artificial and spontaneous fetal mortality rate per 1000 births for each maternal age group and the maternal age-standardized rate for each period and type of household.

Type of Indicators and Households	Maternal Age Group							Maternal Age-Standardized Rate
	15–19	20–24	25–29	30–34	35–39	40–44	45–49	
Artificial fetal mortality rate (Jobless households)								
1995–1999	485.5	215.0	99.2	69.0	88.1	132.6	228.6	113.7
2000–2004	406.4	162.9	80.0	63.7	68.5	104.0	219.4	92.6
2005–2009	316.5	131.9	70.0	56.8	58.7	78.8	131.0	78.8
2010–2014	272.2	108.4	55.5	43.7	48.7	63.0	125.0	63.2
2015–2019	239.7	103.9	54.4	38.5	37.6	60.8	138.6	58.8
Artificial fetal mortality rate (Working households)								
1995–1999	146.0	25.8	7.5	6.6	16.8	73.8	245.0	14.2
2000–2004	143.5	26.7	8.1	6.9	13.3	49.0	206.9	14.0
2005–2009	116.0	24.6	7.9	6.2	10.4	30.0	113.7	12.4
2010–2014	119.5	22.7	7.1	5.4	8.4	20.2	62.3	11.2
2015–2019	106.9	24.4	6.7	5.2	7.6	17.2	53.0	10.9
Spontaneous fetal mortality rate (Jobless households)								
1995–1999	72.3	45.4	34.8	34.3	44.2	64.8	74.3	38.0
2000–2004	47.1	28.6	22.6	21.7	29.1	43.5	32.3	24.4
2005–2009	29.9	20.9	18.9	21.2	26.9	32.7	59.5	21.0
2010–2014	21.1	16.1	15.7	16.5	23.1	32.5	52.4	16.9
2015–2019	16.7	12.9	13.9	15.3	19.7	27.5	34.7	14.9
Spontaneous fetal mortality rate (Working households)								
1995–1999	28.2	13.4	11.2	12.9	19.5	39.3	85.7	13.5
2000–2004	21.7	11.5	10.1	11.7	16.6	28.9	62.0	11.9
2005–2009	15.2	10.2	9.3	10.5	14.4	23.6	37.8	10.6
2010–2014	13.7	9.0	8.4	9.6	12.8	20.3	32.1	9.5
2015–2019	11.9	8.2	7.6	9.1	12.2	17.5	27.5	8.8

## Data Availability

Publicly available datasets were analyzed in this study. This data can be found here: [The Vital Statistics: https://www.e-stat.go.jp/stat-search/files?page=1&toukei=00450011 (accessed on 30 April 2021), Statistics Data of Cabinet Office: https://www.esri.cao.go.jp/jp/sna/data/data_list/kakuhou/files/2019/2019_kaku_top.html (accessed on 30 April 2021), The Survey Of Population, Demographics, And Household Number Based On The Basic Resident Register: https://www.e-stat.go.jp/stat-search/files?page=1&toukei=00200241&tstat=000001039591 (accessed on 30 April 2021).

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
