# Peer review of "An Analysis of the Trend of Fetal Mortality Rates among Working and Jobless Households in Japan, 1995–2019"

_ijerph, 2021, doi:10.3390/ijerph18094810_

Round 1
Reviewer 1 Report
This study analyzed the trend of artificial and spontaneous fetal mortality rates in Japan from 1995 to 2019 and also compared the trends between working households and jobless households. This brief report applied age-period-cohort models and showed household differences. Overall, the results are interesting, but the writing and explanation should be improved. I’d like to provide the following comments for the improvement of this report.
Overall:
(1) Extensive editing of English language is required. For example, in the first paragraph of the Introduction, the authors used “although” in consecutive sentences. The key message of each sentence is not clear. Another example in the same paragraph is unspecific terms, such as “these indicators.” The authors didn’t define “these indicators.”
(2) References should be updated and fit the text. Example in the Materials and Methods section: If the authors consider only fetal mortality, the cited references should be more relevant to fetal mortality rather than others like infant mortality. Example in the Discussion section: References #22 and #23 were published in 2000 and 2005, respectively. However, the authors analyzed data between 1995 and 2019. The causes of changes in the trend might be different. I’d like to suggest the author conduct a more comprehensive literature review and update the references.
Results
(3) Page 3, line 105: I did not receive the supplementary materials for review.
(4) Figure 1: I wonder if the vertical axis (or y-axis) titles are correct. Based on the text, the titles of the y-axis look more like mortality rate rather than mortality rate ratio.
(5) Figure 2: The maximum scale values of the vertical axis should be determined because the lines and confidence intervals exceed the value “6.” If the risk ratio equals 1 is the reference, a reference line can enhance the readability.
Author Response
- To the comment “Extensive editing of English language is required. For example, in the first paragraph of the Introduction, the authors used “although” in consecutive sentences. The key message of each sentence is not clear. Another example in the same paragraph is unspecific terms, such as “these indicators.” The authors didn’t define “these indicators.” ‘’
→ I often used “although” in the manuscript, but there were some parts where “although” was inappropriately used. Therefore, I corrected some parts where a connection of sentences was obviously unclear. (Throughout manuscript) Also, I inserted a sentence taking into account the flow of the sentences. (Introduction, first paragraph, lines 6-8).
In addition, I replaced “these indicators” with “representative indicators.” (Introduction, first paragraph, line 11)
If a further correction is needed, the whole manuscript will receive English proofreading again.
- To the comment “References should be updated and fit the text. Example in the Materials and Methods section: If the authors consider only fetal mortality, the cited references should be more relevant to fetal mortality rather than others like infant mortality. Example in the Discussion section: References #22 and #23 were published in 2000 and 2005, respectively. However, the authors analyzed data between 1995 and 2019. The causes of changes in the trend might be different. I’d like to suggest the author conduct a more comprehensive literature review and update the references." →Regarding citation about fetal mortality, I added some references concerning fetal mortality in the Introduction (Introduction, second paragraph, lines 2-4).
Regarding the citation in Discussion, I had cited the references (References #22 and #23) that were relatively old because the number of previous studies concerning induced abortion is relatively few in Japan. However, I newly cited references published in recent years in Discussion (References #32,33,34,43-45,46).
- To the comment “Results(3) Page 3, line 105: I did not receive the supplementary materials for review."
→I planned to submit Table 1 as a supplementary material at first, but changed the plan when submitting the manuscript. Therefore, I deleted the sentence.
- To the comment “Figure 1: I wonder if the vertical axis (or y-axis) titles are correct. Based on the text, the titles of the y-axis look more like mortality rate rather than mortality rate ratio."
→ “Mortality rate ratio” is correct as the label of y-axis. In an age-period-cohort analysis of mortality data, mortality rate ratios within age groups, periods, and cohorts are usually shown. If a mortality rate ratio of an age group is high, it means that mortality rate of the age group is high.
- To the comment “Figure 2: The maximum scale values of the vertical axis should be determined because the lines and confidence intervals exceed the value “6.” If the risk ratio equals 1 is the reference, a reference line can enhance the readability."
→I added a maximal value of y-axis in the upper part of Figure 2. In addition, a reference line was added in Figure 2. (Figure 2)
Reviewer 2 Report
Dear Editor and author,
I’m pleased to review this interesting article to investigate the fetal mortality in Japan. The author found the status of employment may contribute to fetal mortality. I have some question.
- The role of socioeconomic status is an important issue in your study. It’s appreciated to highlight some important events, such as World War or financial crisis of 2007-2008. Furthermore, it’s also helpful to plot the GDP (gross domestic product) by year. It’s expected there’s reverse correlation between GDP and mortality.
- Multiple factors may contribute to fetal mortality. Was the information of gestation available in the Vital Statistics? Did fetus with senior brothers/sisters have higher mortality?
- What’s the definition of jobless of household? Did you both parents were not employed or only mother?
- It seemed you exclude mothers younger than 15 years and older than 50 years. It’s reasonable but more description is clearer.
- How did you calculate twin pregnancy?
- The footnote of figure 2 should be “95%”?
- Table 1 showed the numbers and Table 2 showed the rate per 1000 births. It’s helpful to add total pregnancy numbers in Table 1. It’s clearer.
- The approval of ethical committee should be added.
Thank you!
Author Response
- To the comment “The role of socioeconomic status is an important issue in your study. It’s appreciated to highlight some important events, such as World War or financial crisis of 2007-2008. Furthermore, it’s also helpful to plot the GDP (gross domestic product) by year. It’s expected there’s reverse correlation between GDP and mortality. ‘’
→ Thank you for the suggestion. World Wars occurring after 1995 are generally not mentioned in epidemiological studies in Japan. On the other hand, the financial crisis of 2007-2008 and the Great East Japan earthquake in 2011 are sometimes mentioned also in Japan, and we mentioned these events in Discussion (Discussion, second paragraph, red colored parts). In addition, we showed the values of yearly GDP per capita and age-standardized mortality rate in the supplementary material (Results, third paragraph). There was no reverse correlation in Japan.
- To the comment “Multiple factors may contribute to fetal mortality. Was the information of gestation available in the Vital Statistics? Did fetus with senior brothers/sisters have higher mortality?"
→Although information on gestational-age were available in some of the Vital Statistics data, we could not obtain the data of fetal mortality by gestational-age for each age group and type of household. Similarly, we could not verify whether fetus with senior brothers/sisters have higher mortality or not from the data that are publicly available. It was shown that these factors affect stillbirths in a previous study in Japan. We mentioned about gestational-age and experience of past births in the limitation (Discussion, the last paragraph, lines 5-7).
- To the comment “What’s the definition of jobless of household? Did you both parents were not employed or only mother?"
→It means that none of persons in the household is working. We added the description in the Methods (Materials and Methods, second paragraph, lines 12-13).
- To the comment “It seemed you exclude mothers younger than 15 years and older than 50 years. It’s reasonable but more description is clearer."
→ We mentioned about this exclusion in the Methods (Materials and Methods, second paragraph, lines 3-5).
- To the comment “How did you calculate twin pregnancy?"
→Thank you for the incisive comment. We asked a person in charge of the Ministry of Health, Labor and Welfare about it, and added a description about it in the Method (Materials and Methods, first paragraph, red colored parts).
- To the comment “The footnote of figure 2 should be “95%”?"
→Yes. A part of the footnote was mistakenly deleted in the edition, and I added the footnotes correctly. (Figure 2)
- To the comment “Table 1 showed the numbers and Table 2 showed the rate per 1000 births. It’s helpful to add total pregnancy numbers in Table 1. It’s clearer."
→We showed the total number of births as Table 1. (Table 1) By the way, we cannot obtain the data on total number of pregnancies.
- To the comment “The approval of ethical committee should be added."
→We added about the approval of ethical committee in the end of the manuscript. (Institutional Review Board Statement before References)
Reviewer 3 Report
Overall the topic of the present study seems to be rather interesting. However, the following points should be followed in order for the paper to improve in quality, benefiting both the journal and the author.
- If possible you could add a little bit of information highlighting the significance of your research (e.g. the situation of Japan regarding the population decline)
- There is extensive repetision of some terms throughout the manuscript
- Since there is only one author you should refer as ''I'' instead of ''We''
- Further comments regarding the manuscript can be found in the attached PDF.

Author Response
- To the comment “If possible you could add a little bit of information highlighting the significance of your research (e.g. the situation of Japan regarding the population decline)‘’
→ As you pointed out, this study is associated with the problem of decrease in population in Japan, and I added some descriptions in Discussion. (Discussion, third paragraph)
- To the comment “There is extensive repetision of some terms throughout the manuscript"
→I checked the manuscript again, and I found that I repeatedly used “although”, “Vital Statistics”, and “APC”. I replaced them with other words or deleted them as much as possible. (Throughout the manuscript, particularly the Introduction)
- To the comment “Since there is only one author you should refer as ''I'' instead of ''We''"
→The sentences involving “We” were replaced with sentences using passive expressions throughout the manuscript.
- To the comment “Further comments regarding the manuscript can be found in the attached PDF. "
→
- I corrected the spell of mortality throughout the manuscript.
- To the comment “Results :Please include a general sentence regarding the main idea of the results before presenting them in detail. " →I included the main idea in the beginning for each of the result. (Results)
- To the comment “Please elaborate. (Table 1)" →I added some explanations for Table 1. (Results, first paragraph)
- To the comment “Figure 1&2 To make this figure a little more well presented, it is suggested that you include the x-axes labels only once. Same for figure 2." →Do you mean the x-axes labels (ages, periods, and cohorts) of the upper parts of the Figures should be omitted? I omitted them. (Figure 1 & 2)
- To the comment “Please make this sentence more comprehensive. “Although it is known that SES is associated with the rate of induced abortion in Japan and other countries [4,20,21], the results of this study demonstrated that the rate ratio of the artificial fetal mortality rate between the households was particularly different depending on ages and cohorts. " →I wanted to say that it was shown in this study that degree of the disparity changed depending on ages, periods, and cohorts. I corrected the sentence. (Discussion, first paragraph, lines 2-5)
- To the comment “In the age group "between" " → I corrected it. (Discussion, first paragraph)
- To the comment “Conclusions If possible include a sentence about where further research should focus." →I included a sentence about where further research should focus in the future. (Conclusions, last sentences)
- I corrected the spell of “difference” in the Conclusions. (Conclusions, first sentence)
Round 2
Reviewer 1 Report
Thanks for the author's response and revision.